# Tibial Tuberosity Advancement Techniques (TTAT): A Systematic Review

**DOI:** 10.3390/ani12162114

**Published:** 2022-08-17

**Authors:** Federica Aragosa, Chiara Caterino, Giovanni Della Valle, Gerardo Fatone

**Affiliations:** Department of Veterinary Medicine and Animal Production, University of Naples “Federico II”, 80137 Naples, Italy

**Keywords:** MMT, TTA rapid, MMP, porous TTA, mTTA, TTA-CF, traditional TTA, TTAT, cranial cruciate ligament rupture

## Abstract

**Simple Summary:**

The aim of this manuscript is to report and critically review the currently available literature about so-called “new generation Tibial Tuberosity Advancement Techniques”. According to PRISMA guidelines, the authors investigated and reported data about preoperative planning, surgical procedure, outcome, and complications of these different techniques. The main problems addressed were the lack of prospective studies with a large study population and univocal data collection about preoperative planning and outcome.

**Abstract:**

(1) Background: Several surgical techniques were described for the treatment of cranial cruciate ligament rupture in dogs. This report aims to critically review the available literature focused on preoperative planning, surgical procedure, follow-up, and complications of cranial cruciate ligament rupture treated by tibial tuberosity advancement techniques; (2) Methods: three bibliographic databases: PubMed, Google Scholar, and Scopus were used for a board search of TTAT (canine OR dog). Five GRADE recommendations according to Grading of Recommendations Assessment, Development and Evaluation and Joanna Briggs Institute Critical Appraisal Checklists were applied to the studies included. Data regarding preoperative planning (a measure of advancement), meniscal disease (meniscectomy, meniscal release, and late meniscal tears), and postoperative patellar tendon angle were recorded. Time frame, outcome, and complications were classified according to Cook’s guidelines; (3) Results: from 471 reports yielded, only 30 met the inclusion criteria. The common tangent method was the most commonly reported measurement technique for preoperative planning. The 40.21% of stifles presented meniscal tears at surgery, while 4.28% suffered late meniscal tears. In short-, mid-and long-term follow-ups examined showed a full/acceptable function was shown in >90% of cases. Among all new generation techniques, minor complications were reported in 33.5% of cases and major complications in 10.67%; (4) Conclusions: Compared to traditional TTA, new generation TTAT resulted effective in the treatment of cranial cruciate ligament failure, showing a lower rate of late meniscal injury but a higher rate of minor complications.

## 1. Introduction

Cranial cruciate ligament rupture (CrCL) is the most frequent cause of osteoarthritis and lameness of the hind limbs in dogs [1,2]. Decision-making regarding surgical treatment of stifle instability is the topic of many concerns and studies [3,4]. Currently, the most common surgical procedures used are tibial osteotomies and, in particular, the Tibial Plateau Leveling Osteotomy (TPLO) and the Tibial Tuberosity Advancement (TTA) [5,6]. The latter was proposed by Montavon and Tepic [6,7], with the aim of neutralising cranial tibiofemoral shear force through displacing tibial tuberosity cranially to reach a patellar tendon angle (PTA) of 90° to the tibial plateau [8,9,10]. Further adaptations of this technique include the modified Maquet technique (MMT) [11], TTA rapid [12], the modified Maquet procedure (MMP) [13], modified Maquet Tibial Tuberosity Advancement (mTTA) [14], tibial tuberosity advancement with cranial fixation (TTA-CF) [15] and porous TTA [16] (Figure 1). These new generation tibial tuberosity advancement techniques (TTAT) involve advancing the tibial tuberosity using saw guides of different shapes and sizes, allowing an incomplete osteotomy of the tibial tuberosity [17]. Although in recent years, advances have been made in preoperative measurement methods, a discrepancy does exist between the desired tibial tuberosity advancement preoperatively measured and the actual advancement of the tibial tuberosity surgically carried out [18,19].

To the best of the authors’ knowledge, no comprehensive review is available of information on the new generation TTAT. Therefore, this manuscript aims to investigate and critically review the scientific evidence about preoperative planning, outcome, and complications by a systematic review of the currently available literature.

## 2. Materials and Methods

### 2.1. Aim and Literature Search Strategy

This systematic literature review followed the PRISMA (preferred reporting items. for systematic reviews and meta-Analyses) flowchart and is in accordance with PRISMA’s statement [20], including the published retrospective or prospective studies of dogs undergoing new generation TTAT for cranial cruciate ligament rupture. Inclusion criteria were a sound description of preoperative planning (measurement of advancement), follow-up, clinical outcome, and complications. To be included, a study had to use defined complications relating to surgery. The studies lacking descriptions of complications were excluded. No language restrictions were applied. Four independent reviewers (CC, GDV, FA, GF) searched the Pubmed, Google Scholar, and Scopus databases from 2011 to May 2022. Finally, we updated the database search on 15 June 2022, examining references cited in study reports included in the systematic review.

Three known relevant studies [21,22,23] were used to identify records within databases. Search terms were also checked using the Pubmed PubReMiner word frequency analysis tool. Candidate search terms were identified by looking at words in those records’ titles, keywords, and abstracts. The search strategy was developed by one of the authors (GF) and validated by testing whether it could identify the three known relevant studies. The electronic search phrases used were “modified Maquet technique” (canine OR dog), “TTA Rapid” (canine OR dog), “modified Maquet procedure” (canine OR dog), “mTTA” or “modified Tibial Tuberosity Advancement” (canine OR dog), “Tibial tuberosity Advancement with Cranial Fixation” or “TTA-CF” (canine OR dog), “Porous TTA” (canine OR dog) for all fields. Moreover, the references list of the papers selected was critically reviewed to improve the sources. A retrospective temporal limitation was placed on 2011 for MMT for relevant publications because Etchepareborde described MMT in 2011 [11]. For all techniques included, the data limit was chosen based on the first description in the literature. (TTA Rapid, 2015 [12]; MMP, 2014 [13]; mTTA, 2016 [14]; TTA-CF [15], 2018; Porous TTA, 2019 [16]). The flow diagram of the literature search and study selection process for MMP is described in Figure 2.

### 2.2. Study Selection

We reviewed titles and abstracts of all records and discussed inconsistencies until consensus was obtained. Then two researchers (FA, CC) independently screened the titles and abstracts of all articles retrieved. In case of disagreement, consensus on which articles to screen full text was reached by discussion. If necessary, a third researcher (GVD) was referred to make the final decision. Next, two researchers (FA, CC) independently screened full texts to determine eligibility. Again, all discrepancies were resolved between the authors, with the availability of a third-party adjudicator (GF).

### 2.3. Data Extraction

Two review authors (FA, CC) independently extracted data from eligible studies. Extracted data were compared, and any discrepancy was resolved through discussion or consulting another researcher (GF). The descriptive variables extracted were the author’s name, study year, sample size, measurement method for required advancement, presence of meniscal injuries, follow-up, and complications. Any measure of planning, outcome, and complications was eligible for inclusion. Results for preoperative planning were reported as a percentage. When recorded, we provided a percentage of dogs with meniscal injuries at presentation undergoing meniscectomy and the percentage of meniscal release performed during TTAT. Furthermore, the number of late meniscal tears was recorded. Postoperative patellar tendon angle (PTA) was reported as a mean for each study and range or standard deviation when provided.

No restrictions on the length of follow-up or number of measurement time points were considered when interpreting study findings. According to Cook et al. (2010) guidelines [24], we summarised complications as catastrophic, major and minor and divided time frames as perioperative (0–3 months), short-term (3–6 months), and mid-term (6–12 months) and long-term (>12 months). If several assessments were performed during a time frame, we considered the last one for analysis. Clinical outcome was assessed as full function, acceptable function and unacceptable function based on restoration of performances from the preinjury period, and we recorded results as a percentage. If the outcome was reported as lameness degree, we merged data for full/acceptable function of dogs with no lameness or sporadic lameness.

We decided to present the results grouped according to the technique used in tables, but we listed results considering every domain. When data were missing or unclear, corresponding studies were excluded from syntheses.

GDV and FA independently assessed the quality of papers included using the Joanna Briggs Institute (JBI) Critical Appraisal Checklists. Every study design was combined with the corresponding JBI checklist. The case report checklist comprised a total of 8 sub-items, and the case series checklist contains 10 sub-items. In this systematic review, we verified case reports and case series checklists to increase the accuracy of the evaluation. If it was clearly described, it was evaluated as “Yes”, “No” if it was not presented, “Unclear”, if it was not clear, and “Not applicable” if it could not be applied. All disagreements were resolved by discussion. In each of the sub-items, the number of studies evaluated as “Yes”, “No”, “Unclear”, and “Not applicable” were reported. According to Munn et al. (2019) guidance [25], we presented the results of critical appraisal for all questions via a table rather than summarising with a score since several study designs were included in this review.

Two authors (FA, CC) independently assessed the certainty of the evidence, using the five GRADE recommendations (study limitations, consistency of effect, imprecision, indirectness, and publication bias) as it related to the studies that contributed data to analyses. We assessed the certainty of evidence as high, moderate, low, or very low [26].

Considering the inclusion of different study designs and interventions, synthesis without meta-analysis (SwiM) checklist [27] was applied. Meta-analyses could not be undertaken due to the heterogeneity of surgeries and study designs.

## 3. Results

### Database Review

We found 471 records in database searching from 2011 to June 2022 (depending on the first publication for each technique). After duplicate removal (n = 117) and exclusion of book sections, ex vivo studies, or thesis (n = 178), we screened 176 records, from which we reviewed 68 full-text documents, and finally included 29 papers [3,11,12,13,15,16,21,22,23,28,29,30,31,32,33,34,35,36,37,38,39,40,41,42,43,44,45,46,47,48]. Later, we searched records from the reference lists of initially included studies, founding one paper that fulfilled inclusion criteria [14]. Only 30 papers met the inclusion criteria and reported a mean of 35 cases (range 1–174). We excluded 33 studies from our review because they treated other surgical techniques than TTAT and 6 studies about cats. Included studies by year, study design, number of cases, GRADE, and surgical planning are summarised in Table 1.

We included the following six techniques in this systematic review: MMT [11], TTA rapid [12], MMP [13], mTTA [14], TTA-CF [15], and porous TTA [16]. A total of 1051 stifles were reviewed: MMT (n = 415), TTA rapid (n = 292), MMP (n = 154), mTTA (n = 59), TTA-CF (n = 25) and porous TTA (n = 106). The modified Maquet technique was applied in 10 studies, while TTA rapid in 8, MMP in 7, mTTA in 3, TTA-CF in 3, and porous TTA in 2. One paper [21] included several surgical techniques (MMT, TTA rapid, MMP, and mTTA), so we listed them in tables for each domain.

Nine studies included have a very low level of certainty (30%), 4 papers have a low level (13.3%), 7 moderate (23.3%), and 10 high levels (33.3%). Thirteen studies have a prospective design (43.33%).

The method used to measure the amount of advancement of the tibial tuberosity was reported in 24/30 studies, with the common tangent method being most represented (n = 8), followed by Orthomed (n = 5). In nine studies, no measurement technique was indicated. Only 6/30 studies reported postoperative PTA (20%). Among the reports considered, 85/108 stifles had postoperative angles clearly defined (78.7%), and postoperative PTA was in the reported range of 90 ± 5°, allowing neutralisation of tibiofemoral shear forces according to Kapler et al. (2015) [22].

A total of 226 cases of meniscal tears detected at surgery time were described in 15 papers, all treated by meniscectomy. In five articles, no meniscal tear was detected during the surgical inspection by arthrotomy or arthroscopy. In 10 papers, no data for meniscal injuries were given. Considering 562 stifles undergoing surgery, which were described in 20 papers, 40.21% presented meniscal tears at surgery time. The meniscal release is reported in 5/30 papers (58 stifles), but it was not performed in 14/30 articles. As regards late meniscal tears, 16/30 papers reported injuries during follow-up time, with 22 cases recorded in 516 stifles (4.28%).

In all papers, recovery was clinically assessed, except for two [22,33], for which only data regarding PTA [22] or radiography [33] were registered. Postoperative radiographic assessments were performed in 27 studies (90%), while an owners’ survey was employed as an outcome assessment in 7 papers (23.3%). Other procedures assessing the follow-up, such as gait analysis (13.3%), evaluation of post-operative stifle range of motion-ROM (3.3%), or baropodometric score (3.3%), were less frequently employed. According to the percentage reported in 21/30 studies (70%), the mean perioperative recovery is 64.76% of full/acceptable function. Perioperative follow-up was reported in 455/495 treated stifles, and a full/acceptable outcome was reported for 340/455 stifles (74.7%). All data extracted regarding outcomes are listed in Table 2.

Short-term follow-up was recorded in 21/30 papers (70%) with a mean of 88.9% of full/acceptable function. These studies examined 648 stifles, but follow-up was reported in 575 stifles in total, 535 of the latter (93%) showed full/acceptable function.

Mid-term follow-up was reported in 15/30 papers (50%) with a mean of 87.8% of full/acceptable function. Follow-up evaluations revealed a full/acceptable function in 313 stifles on 337 examined (92.9%) during this time frame. Papers cited described a total of 390 surgeries, but mid-term follow-up was available in 86.4% of cases.

Five papers collected data about long-term follow-up (16.7%), with a mean of 96.35% of full/acceptable outcomes. Eighty-seven stifles in these studies showed a full/acceptable function upon 92 stifles recorded (94.6%). These papers listed 208 TTATs in total, but only 92 surgeries had a long-term follow-up (44.2%).

Minor complications were recorded for 27/30 papers included (90%), while major complications were listed by all studies except one [11]. No catastrophic complications were reported in any study. Minor and major complications are summarised in Table 3.

The mean minor complication rate was 20.94% for 27 papers. The most frequent minor complications reported were tibial crest fissures and fractures, consisting of 274 of 967 stifles (28.33%) and representing 84.57% of minor complications collected in all studies (274/324). According to reviewed literature, these 70 fractures of the tibial crest and 204 fissures did not need further treatment. For six stifles, minor complications were not clearly defined [14,37].

The mean major complication rate was 24.05% for 29 studies. Considering 1031 were stifles examined, 110 surgeries suffered major complications (10.67%). Most commonly, tibial crest fissures were reported, with 31 cases (28.18% of all major complications listed). Fissures that required further treatment represented 3% of the stifles included. As regards tibial crest fractures, 25/1031 stifles were described (2.43%), followed by 14 surgical site infections (1.36%) and 13 tibial diaphysis fractures (1.26%).

## 4. Discussion

This paper reviews and summarises the available evidence of TTAT in dogs through a systematic review of currently available literature. Although TTA has become increasingly popular, limited evidence was found in the veterinary literature with a sound description of preoperative planning, outcome, and complications. Overall, according to the literature reviewed, a limited number of studies focused on this topic. The data is derived mainly from retrospective studies with a limited number of cases. In addition, only 1/3 of the included papers present a high level of evidence (33.3%), so the extracted data needs critical interpretation.

The debate on the surgical treatment of CrCL is still animated. Despite the high number of procedures developed to stabilise the stifle joint, few studies reported mid- to the long-term outcomes, and even fewer studies reported preoperative planning. As a matter of fact, the predominant form of research was observational case series, resulting in the preponderance of studies with a low level of evidence. Unfortunately, this type of study is limited by confounding variables that decrease the evidentiary value. The only study design that can determine a causal interference is a randomised, controlled clinical trial, and there was only one such study included in this systematic review [28].

Beyond those included in this review, several techniques and adaptations were proposed to achieve joint stabilisation, such as TTA-2, fusion TTA, and circular TTA. However, no evidence was available in the literature for these techniques. Fusion TTA was only anecdotally reported in thesis or conference [49], while for TTA-2, only in vitro study was published [50,51]. For this reason, they were excluded.

Different methods were adopted to assess the advancement required to obtain a postoperative PTA of 90 ± 5°. According to the reviewed literature, 1/3 of papers selected the common tangent method. As previously reported, this method has poor reliability [52] and the tendency to underestimate the necessary advancement [53], leading to under-correction during surgical planning.

Only 20% of studies included recorded postoperative PTA, usually as a mean. According to them, 78.7% of surgeries obtained a final PTA of 90° (±5°), allowing the neutralisation of the shear forces [22,54]. These results could influence the outcome, as a postoperative PTA outside this range could lead to instability and persistent cranial tibial subluxation [52,55]. This residual cranial tibial translation is a potentially post-operative finding after TTAT, contributing to late meniscal tears [56,57]. In agreement with this consideration, we chose to analyse meniscal injuries separately from other complications, considering presentation, meniscectomy, and meniscal release. Among the studies examined, meniscal injury was found in 40.21% of stifles undergoing TTAT and consequently treated with meniscectomy.

Previously published studies reported 10% postoperative secondary meniscus damage after traditional tibial tuberosity advancement [57], these rates can be as high as 20% without meniscal release [58]. Although the release of the medial meniscus disturbs load transmission through the meniscus, increasing instability and cartilage loading [59]. Late meniscal tears in studies included in this review (4.28%) are lower than previously reported. As previously noted, the choice of meniscal release probably influenced the incidence of postoperative meniscal injuries. Still, it is not possible to investigate this result statistically as only five studies chose this technique [12,23,29,41,45].

Nevertheless, all late meniscal tears were recorded in 16 studies where the meniscal release was not performed, except for 1 knee [41]. A possible reason for the relatively high number of secondary meniscal injuries after traditional TTA could be persistent craniocaudal stifle instability [55]. However, considering that few studies evaluated long-term follow-up, probably not all meniscal injuries were identified.

There was various evidence on the clinical outcome for procedures included. This is an expected result because several methodologies and outcome assessments were used in the papers, including visual gait evaluation, owner perceptions, and objective force plate gait analysis. The interpretation and clinical implications may vary even within a singular outcome assessment. To minimise the mistakes in data interpretation, we categorised the outcome using the time frames proposed by Cook et al. (2010) [24].

Most of the reviewed papers provided clinical and radiographic assessments during the follow-up. Unfortunately, for TTAT, other objective assessment tools such as gait analysis or baropodometric score were not routinely employed. Given all surgical techniques, 74.7% of cases regained a full/acceptable limb function in the perioperative period, 93% in short-term follow-up, 92.9% during mid-term and 94.6% up to 1 year (long-term) showing to be able to treat the stifle deficiency over time. Nevertheless, previous studies using gait analysis as an assessment instrument of outcome concluded that despite the significant improvement after traditional TTA, normal limb function was not wholly restored [55,57]. Moreover, comparative studies demonstrated the superiority of TPLO over TTA, achieving a higher level of functional outcome [3,4]. At the same time, a systematic review of techniques for treating CrCL rupture in dogs established the superiority of TPLO over extracapsular sutures but found insufficient evidence concerning differences in the outcomes of TPLO and TTA [60]. Further research comparing TPLO and TTA with subjective gait analysis could not detect differences in the decrease in lameness between the techniques, while objective gait analysis supports the superiority of TPLO [61]. All these papers omitted the new generation of TTA techniques. However, the overall comparisons of these studies are difficult due to the lack of compliance in the data among publications.

To avoid mistakes, complications were classified as catastrophic, major and minor, as defined by Cook and colleagues [24]. In the present study, 324/967 surgeries (33.5%) suffered a minor complication. Tibial tuberosity fractures and fissures were described in 84.57% of cases, but only episodically. These complications required further treatments. These results are not conforming to the minor complication rates reported for traditional TTA. Beer et al. (2018), in a previous systematic review on traditional techniques, recorded 11.6% of minor complications [61]. This evidence cannot be generalised to the new TTAT since the rationale of osteotomy, the spreader and the fixation system employed in these techniques are substantially different from the original TTA, leading to a different risk of complications. Moreover, the major complications reported accounted for 10.67%. Among these, tibial tuberosity fractures and fissures requiring revision surgery accounting for 6%.

Unlike previously published studies, late meniscal injuries requiring second surgery were not counted. Probably, if included, the major complication rate would be higher. Nevertheless, our results align with a previous systematic review of traditional TTA, where major complications accounted for 13.2% [61]. This evidence highlights that the risk of requiring surgical revision is the same for traditional and new-generation TTA.

## 5. Conclusions

The aim of this systematic review is to summarise the currently available literature about new generation TTAT according to PRISMA guidelines. The main limitation is the lack of randomised, controlled clinical trials with a large study population and univocal data collection about preoperative planning, outcome, and complications.

Our results show that the frequency of minor complications is higher than previously reported for traditional TTA. Including recently developed techniques may have conditioned this result, as their experience has not been established over time. Conversely, the incidence of late meniscal tears is lower for new generation TTAT than for traditional TTA, but most of the included studies only recorded data in perioperative and short-term follow-up. Further studies with prospective designs are needed on the new generation of TTAT to support the hypothesis that these techniques could reduce the rate of late meniscal tears.

## Figures and Tables

**Figure 1 animals-12-02114-f001:**
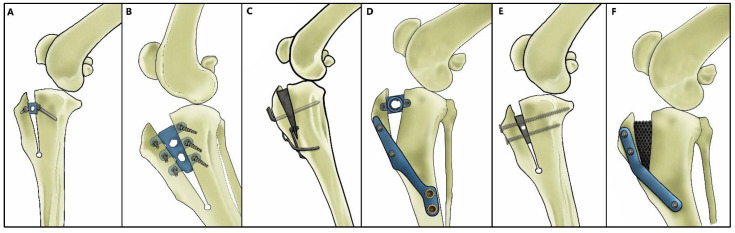
Graphic illustrations of TTAT included: (**A**) modified Maquet technique (MMT), (**B**) TTA rapid, (**C**) modified Maquet procedure (MMP), (**D**) modified Maquet tibial tuberosity advancement (mTTA), (**E**) tibial tuberosity advancement with cranial fixation (TTA-CF) and (**F**) porous TTA (designed by Claudio Palumbo).

**Figure 2 animals-12-02114-f002:**
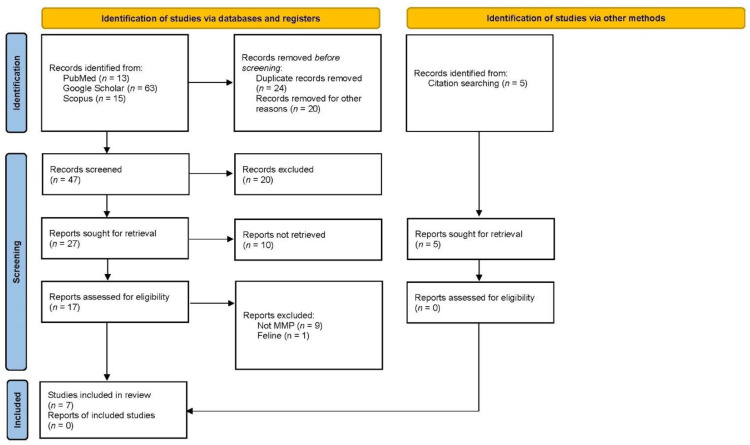
PRISMA 2020 flow diagram for new systematic reviews included searches of databases, registers, and other sources.

**Table 1 animals-12-02114-t001:** List of reviewed papers: study details, sample size, surgical planning, and meniscal injuries.

Authors	Study Design	GRADE	N. Stifles	Surgical Technique	Measurement Technique	Meniscal Tears at Surgery	Meniscectomy	Meniscal Release	Late Meniscal Tears	PTA Post	Year
*Lorenz et al.*	Case series	Very Low	1	MMP	-	1	-	-	-	-	2014
*Kapler et al.*	Retrospective study	Very Low	38	MMP	Orthomed and Modified Tibial Tuberosity Advancement Measurement Method	-	-	-	-	95.9° (86.7°–108.2°)	2015
*Ness et al.*	Clinical trial	Moderate	26	MMP	Orthomed	-	NO	NO	-	-	2016
*Knebel et al.*	Prospective, randomised, controlled study	High	35	MMP	Orthomed	22	22	NO	2	-	2020
*Terreros et al.*	Prospective clinical study	Moderate	15	MMP	Tibial Plateau Method	4	4	4	-	93.4° ± 2.1°	2020
*Della Valle et al.*	Prospective clinical study	Moderate	35	MMP	Orthomed	27	27	6	0	89.7° ± 2.3°	2021
*Serrani et al.*	Retrospective study	Low	4	MMP	-	-	NO	NO	NO	95.75°	2022
*Etchepareborde et al.*	Retrospective study	Very Low	20	MMT	Transparency (Kyon)	8	8	NO	2	-	2011
*Ramirez* et al.	Retrospective study	Moderate	84	MMT	Transparency (Kyon)	39	39	NO	3	-	2015
*Marques et al.*	Case report	Very Low	1	MMT	Orthomed	NO	NO	NO	NO	-	2017
*Marques et al.*	Case series	Very Low	2	MMT	-	NO	NO	NO	NO	-	2017
*Lefebvre et al.*	Retrospective study	Moderate	174	MMT	-	-	-	-	-	-	2017
*Retallack et al.*	Retrospective clinical cohort study	Moderate	35	MMT	-	21	21	14	NO	-	2017
*De Barros et al.*	Prospective clinical study	Very Low	21	MMT	Software?	-	-	-	-	-	2018
*Valino-Cultelli et al.*	Prospective randomised study	High	24	MMT	Tibial Plateau Method	-	-	-	-	-	2021
*Valino-Cultelli et al*.	Prospective clinical study	High	53	MMT	Tibial Plateau Method	-	-	-	-	-	2021
*Serrani et al.*	Case series	Low	1	MMT	Tibial Plateau Method	NO	NO	NO	1	83°	2022
*Samoy et al.*	Prospective clinical study	High	50	TTA Rapid	Common tangent Method	21	21	29	0	-	2015
*Arican et al.*	Prospective study	High	17	TTA Rapid	Template and Common tangent Method	NO	NO	NO	-	-	2017
*Butterworth et al.*	Retrospective study	Moderate	152	TTA Rapid	Tibial Axis Method	44	44	NO	9	-	2017
*Dyall et al.*	Retrospective study	Low	48	TTA Rapid	Anatomical Landmark Method and Common tangent Method	19	19	NO	2	90.8° ± 2.9°	2017
*Heremans et al.*	Case report	Very Low	1	TTA Rapid	-	NO	NO	NO	NO	-	2017
*Livet et al.*	Prospective randomised study	High	13	TTA Rapid	Long Axis Method	4	4	NO	2	91.1° (89.1–92.9°)	2019
*Roydev et al.*	Retrospective study	Low	10	TTA Rapid	Common Tangent Method	5	5	5	1	-	2021
*Serrani et al.*	Retrospective study	Low	1	TTA Rapid	-	1	1	NO	NO	96°	2022
*Trisciuzzi et al.*	Retrospective study	Low	41	Porous TTA	Common Tangent Method and Tibial Plateau Angle Inclination Method	-	-	-	-	-	2019
*Villavicencio et al.*	Prospective study	High	65	Porous TTA	Common Tangent Method	-	-	-	-	-	2020
*Mendeiros et al.*	Prospective clinical study	High	42	mTTA	Transparency	-	-	-	-	-	2016
*Morato et al.*	Prospective study	High	16	mTTA	-	5	5	-	-	-	2019
*Serrani et al.*	Retrospective study	Low	1	mTTA	-	NO	NO	-	-	94°	2022
*Zhalniarovich et al.*	Prospective study	High	22	TTA CF	Common Tangent Method	5	5	NO	NO	-	2018
*Adamiak et al.*	Case report	Very Low	2	TTA CF	Common Tangent Method	NO	NO	NO	NO	-	2018
*Zhalniarovich et al.*	Case report	Very Low	1	TTA CF	-	-	-	-	-	-	2019

**Table 2 animals-12-02114-t002:** The outcome of reviewed papers is divided by surgical technique and time frame.

Authors	N. Stifles	Surgical Technique	Recovery Assessment	Perioperative Recovery	Recovery Short-Term	Recovery Mid-Term	Recovery Long-Term	Year
*Lorenz et al.*	1	MMP	Clinical and Radiographical	0.00%	-	-	-	2014
*Kapler et al.*	48	MMP	PL-TPA	52.60%	-	-	-	2015
*Ness et al.*	26	MMP	Clinical and Radiographical	92.00%	-	84.70%	-	2016
*Knebel et al.*	35	MMP	Clinical, Radiographical, and Gait Analysis	48.40%	77.40%	80.60%	-	2020
*Terreros et al.*	15	MMP	Clinical, Radiographical, and Owners survey	76.92%	-	92.30%	-	2020
*Della Valle et al.*	35	MMP	Clinical, Radiographical, and Gait Analysis	-	54.30%	-	-	2021
*Serrani et al.*	4	MMP	Clinical and Radiographical	0%	-	-	-	2022
*Etchepareborde et al.*	20	MMT	Clinical and Radiographical	80.00%	100.00%	-	-	2011
*Ramirez et al.*	84	MMT	Clinical, Radiographical, and Owners Survey	-	100.00%	-	91.00%	2015
*Marques et al.*	1	MMT	Clinical	100.00%	100.00%	100.00%	-	2017
*Marques et al.*	2	MMT	Clinical and Radiographical	-	100.00%	-	-	2017
*Lefbvre et al.*	174	MMT	Radiographical	-	-	-	-	2017
*Retallack et al.*	35	MMT	Clinical and Radiographical	-	-	-	-	2017
*De Barros et al.*	21	MMT	Clinical and Owners Survey	-	-	81.00%	-	2018
*Valino-Cultelli et al.*	24	MMT	Clinical and Radiographical	72.20%	100.00%	-	-	2021
*Valino-Cultelli et al.*	53	MMT	Clinical and Radiographical	74.30%	97.10%	-	-	2021
*Serrani et al.*	1	MMT	Clinical and Radiographical	0.00%	0.00%	0.00%	0.00%	2022
*Samoy et al.*	50	TTA Rapid	Clinical and Radiographical	-	96.00%	-	-	2015
*Arican et al.*	17	TTA Rapid	Clinical and Radiographical	82.35%	82.35%	-	-	2017
*Butterworth et al.*	152	TTA Rapid	Clinical, Radiographical, and Owners survey	-	99.00%	97.00%	-	2017
*Dyall et al.*	48	TTA Rapid	Clinical, Radiographical, and Owners survey	94.00%	-	95.30%	95.30%	2017
*Heremans et al.*	1	TTA Rapid	Clinical and Radiographical	0.00%	100.00%	100.00%	-	2017
*Livet et al.*	13	TTA Rapid	Clinical, Radiographical, Gait Analysis, and Owners Survey	-	100.00%	100.00%	-	2019
*Roydev et al.*	10	TTA Rapid	Clinical, Radiographical, Gait Analysis, and ROM	70.00%	70.00%	90.00%	100.00%	2021
*Serrani et al.*	1	TTA Rapid	Clinical and Radiographical	0.00%	0.00%	0.00%	0.00%	2022
*Trisciuzzi et al.*	41	Porous TTA	Clinical, Radiographical and Baropodometric score	73.00%	-	100.00%	-	2019
*Villavicencio et al.*	65	Porous TTA	Clinical and Radiographical	87.69%	100.00%	-	-	2020
*Mendeiros et al.*	42	mTTA	Clinical and Radiographical	56.41%	95.00%	-	100.00%	2016
*Morato et al.*	16	mTTA	Clinical and Radiographical	100.00%	100.00%	-	-	2019
*Serrani et al.*	1	mTTA	Clinical and Radiographical	0.00%	0.00%	-	-	2022
*Zhalniarovich et al.*	22	TTA CF	Clinical, Radiographical, and Owners Survey	100.00%	95.45%	95.45%	95.45%	2018
*Adamiak et al.*	2	TTA CF	Clinical and Radiographical	100.00%	100.00%	100.00%	-	2018
*Zhalniarovich et al.*	1	TTA CF	Clinical and Radiographical	0.00%	100.00%	100.00%	-	2019

**Table 3 animals-12-02114-t003:** Complications are expressed in percentage and in detail for papers included.

Authors	N. Stifles	Surgical Technique	Minor Complications	Details	Major Complications	Details	Year
*Lorenz et al.*	1	MMP	-	-	100%	Tibial tuberosity fracture (1)	2014
*Kapler et al.*	48	MMP	-	-	6.25%	Crest fracture (1); tibial fracture (1); Implant motion (1)	2015
*Ness et al.*	26	MMP	15.40%	Cranial displacement of the distal end of the tibial tuberosity (4)	7.70%	Tibial diaphyseal fractures (2)	2016
*Knebel et al.*	35	MMP	-	-	14.30%	Implant removal due to seroma formation (2); implant breakage or loosening (1); tibial fracture (1); wound complications (1)	2020
*Terreros et al.*	15	MMP	60%	Incisional redness (4), cortical hinge fractures (6)	20.00%	Deep (1) and superficial (2) surgical site infections	2020
*Della Valle et al.*	35	MMP	65.70%	Cortical hinge fissures (22); Seroma (1)	8.57%	Surgical site infection (1); tibial tuberosity fracture (2)	2021
*Serrani et al.*	4	MMP	0%	-	100%	Surgical site infection and implant loosening (1); implant complications (1); distal tibial crest fracture (2);	2022
*Etchepareborde et al.*	20	MMT	5.00%	Tibial crest fracture (1)	-	-	2011
*Ramirez et al.*	84	MMT	9.52%	Cortical hinge fissures (5); Bandage soares (2), lameness of unknown origin (1)	30.95%	Cortical hinge fractures (5); cortical hinge fissures (16); fracture of tibial diaphysis (1), wound dehiscence (2), septic arthritis (1); wound secondary to cerclage wire (1)	2015
*Marques et al.*	1	MMT	0.00%	-	0.00%	-	2017
*Marques et al.*	2	MMT	0.00%	-	100.00%	Fissure of tibial crest (1); fracture of tibial crest (1)	2017
*Lefbvre et al.*	174	MMT	39.65%	Fissures (56); fractures of cortical hinge and tuberosity (13)	9.20%	Fissures of cortical hinge (8); Fractures of cortical hinge (6); Fractures tibial shaft (2)	2017
*Retallack et al.*	35	MMT	20.00%	Tibial crest fractures (7)	5.71%	Surgical site infections (2)	2017
*De Barros et al.*	21	MMT	0.00%	-	4.76%	Seroma (1)	2018
*Valino-Cultelli et al.*	24	MMT	8.33%	Fracture of the distal cortical of the tibial crest (1); mass on the incision region (1)	12.50%	Tension band wiring rupture with or without tibial crest displacement (3)	2021
*Valino-Cultelli et al.*	53	MMT	9.40%	Fracture of the distal cortical of the tibial crest (4); mass on the incision region (1)	9.40%	Tension band wiring rupture with or without tibial crest displacement (4); implant rupture (1)	2021
*Serrani et al.*	1	MMT	0.00%	-	0.00%	-	2022
*Samoy et al.*	50	TTA Rapid	32.00%	Thickened patellar ligament (12); fracture of the distal cortex (4)	4.00%	Tibial crest fractures (2)	2015
*Arican et al.*	17	TTA Rapid	25.00%	-	23.50%	Tibial crest fractures (4)	2017
*Butterworth et al.*	152	TTA Rapid	71.05%	Fissures (104); fracture of tibial crest (1); drill bit broke (3);	5.92%	Fissures (3); surgical site infections (3); fractures of tibia surgically treated (3)	2017
*Dyall et al.*	48	TTA Rapid	6.25%	Fissures (2); mild tissue swelling (1)	10.42%	Fissures (2); non-displaced tibial fractures (2); incisional infection (1)	2017
*Heremans et al.*	1	TTA Rapid	0.00%	-	100.00%	Fracture of the tibial tuberosity and a fissure of the proximal tibia (1)	2017
*Livet et al.*	13	TTA Rapid	23.08%	Distal tibial crest fractures (2); implant loosening (1)	23.08%	Patellar desmitis (1); implant loosening (1); distal tibial crest fracture (1)	2019
*Roydev et al.*	10	TTA Rapid	20.00%	Seroma (1); distal tibial fissure (1)	10.00%	Avulsion of tibial crest (1)	2021
*Serrani et al.*	1	TTA Rapid	0.00%	-	0.00%	-	2022
*Trisciuzzi et al.*	41	Porous TTA	19.51%	Fracture of tibial tuberosity (6); surgical wound dehiscence (2)	2.44%	Fracture of tibial tuberosity (1)	2019
*Villavicencio et al.*	65	Porous TTA	66.15%	distal tibial tuberosity fractures (15), of which 2 with avulsion; distal tibial tuberosity fissures (8), of which 1 with avulsion; lameness after trauma or resting (12); superficial infection (3); implant ruptures (2); dermatitis (2); fissure between screws (1)	1.50%	Infection and implant removal (1)	2020
*Mendeiros et al.*	42	mTTA	4.76%	-	9.52%	Suture dehiscence and superficial infection (1); screw loosening and tibial tuberosity displacement (1); implant failures (2)	2016
*Morato et al.*	16	mTTA	37.50%	Fractures of tibial crest (6)	6.25%	Surgical site infection (1)	2019
*Serrani et al.*	1	mTTA	0.00%	-	100.00%	Distal tibial crest fractures (1)	2022
*Zhalniarovich et al.*	22	TTA CF	27.00%	Fissures through Maquet hole (6)	0.00%	-	2018
*Adamiak et al.*	2	TTA CF	0.00%	-	0.00%	-	2018
*Zhalniarovich et al.*	1	TTA CF	0.00%	-	100.00%	Tibial crest and diaphysis fracture (1)	2019

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
