# Peer review of "Tibial Tuberosity Advancement Techniques (TTAT): A Systematic Review"

_animals, 2022, doi:10.3390/ani12162114_

Round 1
Reviewer 1 Report
Dear. the author
The authors reviewed Tibial Tuberosity Advancement Techniques (TTAT). And the authors investigated and reported data about preoperative planning, surgical procedure, outcome and complications of these different techniques between traditional tibial tuberosity advancement and new generation TTAT.
This paper is well organized, and I evaluated the author’s efforts highly, but has some points which should be revised.
I wrote the detail of review in this manuscript (word file).

Author Response
ANSWERS TO REVIWERS’COMMENTS
As corresponding, the authors (AU) thank the Reviewers for the important and valuable comments that helped to improve the quality of the manuscript. The AU have carefully considered every reviewer’s recommendation and the manuscript has been revised according to their suggestions.
Your Sincerely
Dr. Giovanni Della Valle
Reviewer #1:
Dear. the author
The authors reviewed Tibial Tuberosity Advancement Techniques (TTAT). And the authors investigated and reported data about preoperative planning, surgical procedure, outcome and complications of these different techniques between traditional tibial tuberosity advancement and new generation TTAT.
This paper is well organized, and I evaluated the author’s efforts highly, but has some points which should be revised.
I wrote the detail of review in this manuscript (word file).
Line 18
You can change bold letters in ((2) Methods:).
AU: Okay, the manuscript has been accordingly modified. Please see the text (line:18)
Line 29
You can change >90% to > 90%
AU: Okay, the manuscript has been accordingly modified. Please see the text (line:29)
Line 41-48
I think that if you insert some photo and illustration about each Tibial Tuberosity Advancement Techniques (TTAT) for the reader, it is better to understand each TTAT (Tibial Tuberosity Advancement (TTA), Modified Maquet Technique (MMT) , TTA Rapid, the Modified Maquet Procedure MMP) , and modified Maquet Tibial Tuberosity Advancement (mTTA), Tibial Tuber sity Advancement with Cranial Fixation (TTA-CF) and Porous TTA)
AU: Okay, the manuscript was accordingly modified, inserting an original graphic representation of TTAT included. Please see the text (line: 54)
Line 254-266
The author discuss about meniscus damage between traditional tibial tuberosity advancement and new generation TTAT. I wonder that why is low rate of meniscus damage in new generation TTAT. I
think that new Tibial Tuberosity Advancement Techniques (TTAT) and traditional tibial tuberosity advancement is used same principal OP and biomechanical model. You explained low meniscus damage by meniscal release in new Tibial Tuberosity Advancement Techniques (TTAT) with 5 studies.
You need to discuss explain about meniscus damage between traditional tibial tuberosity advancement and new generation TTAT.
AU: Dear Reviewer, as reported in the text, the choice of meniscal release probably influenced the incidence of postoperative meniscal injuries. Still, it is not possible to investigate this result statistically as only 5 studies chose this technique. (Line:259-261).
We are completely in agreement with you about the foundational biomechanical model is the same in TTAT and Traditional TTA, but, the osteotomy shape and the advancement direction of latter technique are quite different and can predispose to a high risk of residual craniocaudal stifle instability. As reported in the text, a possible reason for the relatively high number of secondary meniscal injuries after traditional TTA could be persistent craniocaudal stifle instability. But considering that few studies evaluated long-term follow-up, probably not all meniscal injuries were identified. Line: (263-266).
Line 258-261
You are able to insert References about 5 studies.
AU: Okay, the manuscript has been accordingly modified. Please see the text (line: 266)
Reviewer 2 Report
Check bibliography: Valiño-Cultelli V, Varela-López Ó , González-Cantalapiedra A, Muszy´nski S, Muszy´nski M, Tomaszewska E. Does PRGF 404 Work? A Prospective Clinical Study in Dogs with A Novel Polylactic Acid Scaffold Injected with PRGF Using the Modified 405 Maquet Technique. Published online 2021. doi:10.3390/ani11082404
Author Response
ANSWERS TO REVIWERS’COMMENTS
As corresponding, the authors (AU) thank the Reviewers for the important and valuable comments that helped to improve the quality of the manuscript. The AU have carefully considered every reviewer’s recommendation and the manuscript has been revised according to their suggestions.
Your Sincerely
Dr. Giovanni Della Valle
Reviewer #2:
Check bibliography: Valiño-Cultelli V, Varela-López Ó , González-Cantalapiedra A, Muszy´nski S, Muszy´nski M, Tomaszewska E. Does PRGF 404 Work? A Prospective Clinical Study in Dogs with A Novel Polylactic Acid Scaffold Injected with PRGF Using the Modified 405 Maquet Technique. Published online 2021. doi:10.3390/ani11082404
AU: Okay, the manuscript has been accordingly modified. Please see the text (line: 441; ref. 36)